# Glycation of Tie-2 Inhibits Angiopoietin-1 Signaling Activation and Angiopoietin-1-Induced Angiogenesis

**DOI:** 10.3390/ijms23137137

**Published:** 2022-06-27

**Authors:** Haiyan Zhou, Tangting Chen, Yongjie Li, Jingcan You, Xin Deng, Ni Chen, Tian Li, Youkun Zheng, Rong Li, Mao Luo, Jianbo Wu, Liqun Wang

**Affiliations:** 1Drug Discovery Research Center, Southwest Medical University, Luzhou 646000, China; zhy818ella@163.com (H.Z.); sbyniso@163.com (Y.L.); youjc15922801974@163.com (J.Y.); xindeng1988@163.com (X.D.); chenniluyi@126.com (N.C.); litianforever@163.com (T.L.); zyktmx@swmu.edu.cn (Y.Z.); hubeilirong@126.com (R.L.); luomao20050908@swmu.edu.cn (M.L.); 2Laboratory for Cardiovascular Pharmacology of Department of Pharmacology, The School of Pharmacy, Southwest Medical University, Luzhou 646000, China; 3Key Laboratory of Medical Electrophysiology, Ministry of Education and Medical Electrophysiological Key Laboratory of Sichuan Province, Collaborative Innovation Center for Prevention and Treatment of Cardiovascular Disease of Sichuan Province, Institute of Cardiovascular Research, Southwest Medical University, Luzhou 646000, China; ctt@swmu.edu.cn

**Keywords:** angiopoietin-1, angiogenesis, endothelial cells, methylglyoxal, Tie-2

## Abstract

The impairment of the angiopoietin-1 (Ang-1)/Tie-2 signaling pathway has been thought to play a critical role in diabetic complications. However, the underlying mechanisms remain unclear. The present study aims to investigate the effects of Tie-2 glycation on Ang-1 signaling activation and Ang-1-induced angiogenesis. We identified that Tie-2 was modified by advanced glycation end products (AGEs) in aortae derived from high fat diet (HFD)-fed mice and in methylglyoxal (MGO)-treated human umbilical vein endothelial cells (HUVECs). MGO-induced Tie-2 glycation significantly inhibited Ang-1-evoked Tie-2 and Akt phosphorylation and Ang-1-regulated endothelial cell migration and tube formation, whereas the blockade of AGE formation by aminoguanidine remarkably rescued Ang-1 signaling activation and Ang-1-induced angiogenesis in vitro. Furthermore, MGO treatment markedly increased AGE cross-linking of Tie-2 in cultured aortae ex vivo and MGO-induced Tie-2 glycation also significantly decreased Ang-1-induced vessel outgrow from aortic rings. Collectively, these data suggest that Tie-2 may be modified by AGEs in diabetes mellitus and that Tie-2 glycation inhibits Ang-1 signaling activation and Ang-1-induced angiogenesis. This may provide a novel mechanism for Ang-1/Tie-2 signal dysfunction and angiogenesis failure in diabetic ischaemic diseases.

## 1. Introduction

Angiogenesis impairment resulting in microvascular insufficiency contributes to the pathogenesis of various cardiovascular complications in diabetes mellitus [1,2] and the underlying mechanisms involved are poorly illustrated. The specific receptor tyrosine kinases (RTKs) expressed on the endothelial cell surface, including those of vascular endothelial growth factor (VEGF) receptor-2 (VEGFR-2) and Tie-2, are critical for angiogenesis [3,4]. However, diabetic patients and diabetic animal models often exhibit defective VEGF/VEGFR-2 [5,6,7] and angiopoietin-1 (Ang-1)/Tie-2 [8,9] signaling transduction, which has been demonstrated to play an important role in diabetes-related impairment of angiogenesis. Our previous studies have shown that the glycation of an extracellular matrix may be involved in the disruption of VEGF/VEGFR-2 angiogenic signaling in endothelial cells [7,10]. Nevertheless, the molecular mechanisms underlying impaired activation of Ang-1/Tie-2 signaling in diabetes have not been fully clarified.

Tie-2 is selectively expressed in endothelial cells and gene deficiency of Tie-2 leads to embryonic lethality due to abnormal vascular development, indicating its indispensable role in maintaining normal vascular function [11]. Ang-1, which is secreted from perivascular cells, has been identified as the major ligand for Tie-2. Binding of Ang-1 to Tie-2 elicits different downstream signaling pathways mainly regulated by Akt, which induces pro-angiogenic responses including the promotion of endothelial cell migration, tube formation and survival [12,13]. Therefore, diabetes-related changes in the structure and function of Tie-2 may bring negative effects on Ang-1/Tie-2 signaling activation and Ang-1-induced angiogenesis.

Methylglyoxal (MGO), a highly reactive metabolism of glucose, significantly increases under hyperglycemic condition in diabetes [14,15]. The increased MGO not only directly impairs cell functions, but also reacts with amino acid residues in proteins to accelerate advanced glycation end product (AGE) formation [16,17]. AGEs formed on proteins in circulation activate receptors for AGEs (RAGE) and subsequently trigger oxidative stress, cell apoptosis and inflammation responses which contribute to the development of diabetic complications [18]. Moreover, cellular proteins may be directly modified by MGO, causing structural and functional abnormalities and it has been shown that glycation alters the functions of cellular proteins, including heat shock protein 27 [19], p38 [20], epidermal growth factor receptor [21], insulin receptor [22,23] and platelet-derived growth factor receptor-β (PDGFR-β) [24]. However, no previous study has identified Tie-2 glycation in diabetes and investigated the effects of this glycation on Ang-1 signaling pathways.

Therefore, in this study, we determined the glycation of Tie-2 in aortae derived from diabetic mice and further tested the hypothesis that MGO-induced Tie-2 glycation may inhibit Ang-1 signaling and Ang-1-induced angiogenesis.

## 2. Results

### 2.1. Tie-2 Is Glycated in HFD-Fed Mice

Firstly, the modification of Tie-2 was investigated in a pathological model in vivo. Specifically, 8-week-old male C57BL/6J mice were fed with a normal chow diet (ND) or a high-fat chow diet (HFD) for 16 weeks, which produced obesity and hyperglycemia (Appendix A). The aortae derived from ND-fed and HFD-fed mice were isolated, fixed, embedded in paraffin, and sectioned. The Tie-2 and AGEs were then immunohistochemically stained in two serial aortic sections and the results demonstrated markedly increased AGE formation in aortae and periaortic tissues derived from HFD-induced obese mice (Figure 1a–c). Moreover, although the expression of Tie-2 showed no significant change (Figure 1a,b,d), the aortae and the small vessels in periaortic tissues from HFD-fed mice yielded a remarkable Tie-2–AGEs colocalization (Figure 1b), indicating that Tie-2 may be modified by AGEs in HFD-treated mice.

### 2.2. Tie-2 Is Glycated in MGO-Treated Human Umbilical Vein Endothelial Cells (HUVECs) In Vitro

MGO, which has been confirmed to significantly increase in diabetes, is known to react with arginine and lysine residues of proteins to form AGE–protein adducts, resulting in impairment of protein function. Therefore, we hypothesized that MGO may induce Tie-2 glycation. To test this hypothesis, HUVECs were exposed to MGO (100 μM) or vehicle control for 24 h and the effects of MGO on Tie-2 modification were determined. The cell lysates were immunoprecipitated with anti-Tie-2 antibody and blotted with anti-AGE antibody. The results showed that incubation with MGO significantly increased the formation of AGEs on Tie-2 immunoprecipitates (Figure 2a), providing evidence for AGE cross-linking of Tie-2 induced by MGO. Furthermore, whether MGO altered the expression of Tie-2 was also detected. HUVECs were stimulated with MGO (0, 50, 100 and 200 μM) for 24 h and the results showed no notable change in total Tie-2 expression (Figure 2b). The membrane proteins were then separated and the Tie-2 expressed at the cell surface was determined. Under the condition used, exposure of HUVECs to MGO (100 μM) for 24 h elicited no significant difference in membrane Tie-2 expression (Figure 2c). Altogether, these results indicate that MGO does not quantitatively alter Tie-2, but induces the glycation of Tie-2 which may lead to Tie-2 dysfunction.

### 2.3. MGO-Induced Tie-2 Glycation Impairs Ang-1 Signaling Activation and Ang-1 Induced Angiogenesis In Vitro

To investigate the impacts of MGO-induced glycation on Tie-2 function, the activation of Ang-1 signaling pathways was determined. HUVECs were exposed to MGO (100 μM) or vehicle control for 24 h, followed by stimulation with Ang-1 (200 ng/mL) for 15 min. Cell lysates were then prepared, separated by SDS-PAGE and analyzed with immunoblotting. Ang-1 significantly increased the phosphorylation of Tie-2 in HUVECs without MGO stimulation, whereas pretreatment with MGO for 24 h inhibited Ang-1-induced Tie-2 phosphorylation (Figure 3a). The downstream signaling activated by Ang-1/Tie-2, such as Akt, was also measured and MGO remarkably inhibited Ang-1-evoked Akt phosphorylation (Figure 3a). These data indicate the MGO-induced modification of Tie-2 may result in Tie-2 dysfunction.

To further investigate the effects of Tie-2 glycation on Ang-1-activated endothelial cell physiological responses, HUVECs pretreated with MGO or vehicle control were seeded onto the upper chambers of a transwell, followed by exposure to Ang-1 (200 ng/mL). After 12 h, cells that had migrated to the lower chambers were stained and counted. Ang-1 significantly increased the migration of endothelial cells without MGO pre-incubation. However, pretreatment with MGO inhibited Ang-1-induced cell migration (Figure 3b). In agreement with the transwell migration assay, MGO also remarkably inhibited cell migration areas stimulated by Ang-1 in the scratch wound healing experiments (Appendix A). To determine the effects of MGO-induced Tie-2 glycation on Ang-1-induced tube formation, HUVECs pre-incubated with MGO or vehicle control were seeded onto a growth factor-reduced Matrigel matrix and exposed to Ang-1 for 12 h. The results also demonstrated that MGO significantly decreased Ang-1-indcued endothelial cell tube formation (Figure 3c). Taken together, all of these data suggest that Tie-2 glycation impairs Ang-1 signaling pathway activation and Ang-1-regulated angiogenesis in vitro.

### 2.4. The Inhibition of Tie-2 Glycation Rescues Ang-1 Signaling Activation and Ang-1-Induced Angiogenesis In Vitro

To clarify the relationship between Tie-2 glycation and Ang-1/Tie-2 signaling dysfunction, the effects of aminoguanidine (AG) (AGE formation inhibitor), which has been demonstrated to prevent MGO-induced protein glycation [21,24], on Ang-1/Tie-2 signaling pathway activation and Ang-1-induced angiogenesis were determined in vitro. HUVECs were pretreated with AG (100 μM) for 1 h and then exposed to MGO (100 μM) for 24 h, followed by stimulation with Ang-1 (200 ng/mL) for 15 min. The Western blotting results demonstrated that the inhibitory effects of MGO on Tie-2 and Akt phosphorylation were significantly prevented by AG pretreatment (Figure 4a). The experiments involving the transwell migration assay and Matrigel tube formation assay also showed AG markedly rescued Ang-1-induced angiogenesis in vitro (Figure 4b,c). Taken together, these data revealed that the inhibition of AGE formation prevents MGO-induced Tie-2 dysfunction, indicating MGO-induced Tie-2 glycation may consequently induce disruption of the Ang-1/Tie-2 signaling pathway.

### 2.5. Glycation of Tie-2 Inhibits Ang-1-Induced Vessel Outgrowth Ex Vivo

To further detect the effects of Tie-2 glycation on Ang-1-induced angiogenesis ex vivo, the aortae isolated from C57BL/6J mice were pretreated with AG (100 μM) for 1 h, followed by exposure to MGO (100 μM) for 24 h. The segments of MGO-treated aortae were then cultured in rat tail collagen I gel in the presence or absence of Ang-1 (200 ng/mL) for 14 d. The histologic analysis revealed a remarkable Tie-2–AGEs colocalization in MGO-treated aortae, which were significantly inhibited by AG pretreatment (Figure 5a), providing evidence for AGE cross-linking of Tie-2 in cultured aortae ex vivo. Moreover, as shown in Figure 5b, Ang-1-induced microvessel outgrowth was significantly impaired in MGO-treated aortae compared to the vehicle control, whereas the blockade of Tie-2 glycation by AG pretreatment rescued the inhibitory effects of MGO on Ang-1-induced vessel sprouting, which is in good agreement with the migration and tube formation results obtained in vitro (Figure 4b,c). These results suggest that MGO-induced Tie-2 glycation plays a key role in impaired Ang-1 signaling and Ang-1-induced angiogenesis ex vivo.

## 3. Discussion

Previous studies suggest that the impairment of the Ang-1/Tie-2 signaling pathway has been proposed to be implicated in the pathogenesis of diabetic microangiopathy [8,9]. However, what leads to Ang-1/Tie-2 signaling dysfunction and the molecular mechanisms involved have not been fully elucidated. In the present study, we firstly identified the glycation of Tie-2 in HFD-treated mice and MGO-treated endothelial cells and further investigated the effects of glycation on Ang-1/Tie-2 signaling and Ang-1-induced angiogenesis. Our results indicate that the direct modification of Tie-2 by AGEs may contribute to impairment of Ang-1/Tie-2 signaling.

During diabetes mellitus, increased AGEs are well known to be implicated in diabetic microvascular complications through inducing oxidative stress, inflammation and cell apoptosis [18]. More remarkably, modification of proteins by AGEs has been shown as one of the critical pathogenic events in diabetes, which probably directly disturbs the structure and functions of target proteins [25,26,27]. Our experiments involving immunohistochemical staining identified significant Tie-2–AGEs colocalization in HFD-treated mice, indicating the direct modification of Tie-2 by AGEs. Although few previous studies have investigated the glycation of Tie-2, a large number of proteins have been verified to be glycated in diabetes mellitus in vivo, including vitronectin [7], fibronectin [10], PDGFR-β [24], and so on, which is at least partly in agreement with the findings in the present study. To model Tie-2 glycation in vitro, MGO, which has been shown to significantly increase in diabetic serum, was used to stimulate HUVECs. Consistent with the in vivo results, our immunoprecipitation experiments also demonstrated Tie-2–AGEs complex formation in MGO-stimulated endothelial cells, which further confirmed Tie-2 glycation. In good agreement with our results, previous studies have demonstrated that MGO induced modification of PDGFR-β by AGEs in mesenchymal cells, smooth muscle cells and fibroblasts [24] and MGO-induced glycation of epidermal growth factor receptor in ECV304 cells [21].

To clarify the significance of Tie-2 glycation in Ang-1/Tie-2 signaling, MGO-treated HUVECs were stimulated with Ang-1. Our results showed remarkably decreased Tie-2 phosphorylation and downstream signaling activation in MGO-treated endothelial cells compared to cells stimulated with vehicle control. Previous studies also showed Ang-1 signaling impairment in mouse heart microvascular endothelial cells cultured in high glucose [9], which is partly consistent with our data. In consideration of the experiments involving AG, which displayed the significant role of the blockade of AGE formation in rescuing Ang-1-induced Tie-2 and Akt phosphorylation, we confirmed the serious impairment of MGO-induced Tie-2 glycation in Ang-1 signaling. This inhibitory effect may result from two mechanisms. On the one hand, MGO-induced glycation of protein amino acid residues can lead to loss of charge and distortion in protein structure, which is associated with decreased ligand-receptor interaction [24,28,29]. On the other hand, protein modification by MGO can also directly reduce the activity of tyrosine kinase [21,23,24,30]. It may be worth noting that the loss of Tie-2 was excluded, since neither total Tie-2 expression nor expression of Tie-2 at the plasma membrane significantly changed.

The attenuated Ang-1/Tie-2 signal transduction has been demonstrated as one of the important mechanisms underlying angiogenesis dysfunction in diabetic complications. Experiments involving cell migration and tube formation in vitro and aortic ring angiogenesis ex vivo displayed that MGO treatment significantly decreased Ang-1-induced cell migration, tube formation and vessel sprouting, whereas the inhibition of AGE formation by AG eliminated the inhibitory effects of MGO on Ang-1 angiogenic responses, further indicating the important role of Tie-2 glycation on dysfunction of Ang-1 signal transduction. Consistent with our results, previous studies also confirmed that Ang-1-induced aortic ring vessel outgrowth was greatly decreased in *db*/*db* mice compared to C57BL/6J mice [8]. However, a limitation of our ex vivo experiments was that we did not prove the inhibition of Tie-2 and Akt phosphorylation. Additional experiments involving immunohistochemical staining of phosphorylated Tie-2 and Akt in aortic rings have to be further performed to confirm the findings. Nevertheless, our ex vivo data complement and support our in vitro results, which demonstrated that Tie-2 glycation inhibits Ang-1 signaling activation and Ang-1-induced angiogenesis. Additional in vivo studies will be needed to further dissect and better clarify the significance of our newly reported regulatory effects of Tie-2 glycation on Ang-1 signaling in other disease models and diabetic patients.

In summary, our present study has found that Tie-2 is glycated in HFD-treated mice and in MGO-treated endothelial cells and the glycation of Tie-2 inhibits Ang-1 signaling activation and Ang-1-induced angiogenesis (Figure 6). The findings indicate that the modification of Tie-2 by AGEs may be involved in the underlying mechanisms of Ang-1/Tie-2 signaling impairment and diabetic microangiopathy. Considering that more and more pharmacological approaches [31,32] with reduction in AGE formation, suppression of RAGE and treatment of insulin resistance have been demonstrated to prevent and treat diabetes and diabetic complications recently, our data may imply a potential target.

## 4. Materials and Methods

### 4.1. Chemicals and Reagents

Primary HUVECs (Catalog No.: 8000) and the endothelial cell medium (Catalog No.: 1001) were obtained from ScienCell Research Laboratories (Carlsbad, CA, USA). Recombinant human Ang-1 (Catalog No.: 923-AN) was obtained from R&D systems (Minneapolis, MN, USA). MGO (Catalog No.: M0252) and AG (Catalog No.: 396494) were purchased from Sigma-Aldrich (St. Louis, MO, USA). Growth-factor-reduced Matrigel matrix (Catalog No.: 356234) was purchased from BD Biosciences (San Jose, CA, USA). Collagen I (Catalog No.: 354236) was obtained from Corning (Corning, NY, USA). Antibodies against phosphorylated Tie-2 (Catalog No.: ab151704) and AGEs (Catalog No.: ab23722) were purchased from Abcam (Cambridge, MG, USA). Antibodies against phosphorylated Akt (Catalog No.: 4060), total Akt (Catalog No.: 4691) and total Tie-2 (Catalog No.: 4224) were obtained from Cell Signaling Technology (Beverly, MA, USA). Anti-Tie-2 (Catalog No.: bs-1300R) and anti-AGEs (Catalog No.: bs-1158R) antibodies used for immunohistochemical staining were purchased from Bioss (Beijing, China). Pierce classic immunoprecipitation kit (Catalog No.: 26146) was obtained from Thermo Scientific (Rockford, IL, USA). The high-fat diet (TP2330055A) and the standard chow diet (TP2330055AC) were obtained from Trophic Animal Feed High-tech Co., Ltd. (Nantong, China).

### 4.2. Animals

C57BL/6J mice (6–8 weeks old) were purchased from the Chongqing Medical University Animal Center (Chongqing China). These mice were maintained in a controlled environment (20–22 °C; 12:12 h light/dark cycles). All procedures for animal use were approved by the Animal Care and Use Committee of Southwest Medical University (approval number: 20200304-015).

### 4.3. HFD-Fed Mouse Model

The 8-week-old male C57BL/6J mice were fed with a HFD (fat 60%, carbohydrate 25% and protein 15%) for 16 weeks, as described previously [33,34]. Age-matched male mice fed with a normal chow diet (fat 10%, carbohydrate 75% and protein 15%) were used as controls.

### 4.4. Histological Analysis

Mouse aortae were isolated from HFD-fed and ND-treated mice, fixed, embedded in paraffin, and sectioned. The 4 μm cross-sections were then prepared and incubated with an antibody to Tie-2 (1:100) or AGEs (1:100) at 4 °C overnight, followed by exposure to horseradish peroxidase-conjugated secondary antibodies for 60 min at 37 °C. The immune complexes were determined using diaminobenzidine substrate working solution.

### 4.5. Cell Culture

HUVECs were cultured in endothelial cell medium with 5% (*v*/*v*) fetal bovine serum (FBS), 1% (*v*/*v*) endothelial cell growth supplement and 1% (*v*/*v*) antibiotic solution (penicillin/streptomycin) [7,10]. When the cells were stimulated with MGO or AG, the endothelial cell medium contained 1% FBS, and when the cells were exposed to Ang-1, FBS-free medium was used. Cells used were passaged 3–10 times.

### 4.6. Cell Scratch Wound Healing Assay

HUVECs were seeded onto 24-well plates, allowed to grow to 90% confluence and exposed to MGO (100 μM) or vehicle control (in this study, phosphate buffer saline was used as the vehicle control unless otherwise indicated) for 24 h. The cells were then scratched with a 200 μL pipette tip, followed by stimulation with Ang-1 (200 ng/mL) or vehicle control for 24 h. Photomicrographs were taken immediately after the scratch and after Ang-1 stimulation. Image J software was used to measure the change in scratch area over time.

### 4.7. Cell Migration Assay

HUVEC migration was also performed using transwell chambers with an 8.0 μm-sized porous membrane. HUVECs cultured in 6-well plates were allowed to grow to 90% confluence and exposed to MGO (100 μM) or vehicle control for 24 h. In some experiments, the cells were pretreated with AG (100 μM) for 1 h before MGO treatment [35,36]. The cells were then detached and 2 × 10^4^ cells were added to the upper chambers, followed by stimulation with Ang-1 (200 ng/mL). After 12 h, cells remaining in the upper chambers were removed and the cells that had migrated to the lower chambers were fixed with 4% paraformaldehyde, stained with 0.25% crystal violet, photographed with a microscope and counted.

### 4.8. Tube Formation Assay

HUVECs cultured in 6-well plates were allowed to grow to 90% confluence and exposed to MGO (100 μM) or vehicle control for 24 h. In some experiments, the cells were pretreated with AG (100 μM) for 1 h before MGO treatment. The cells were then detached and seeded onto 24-well plates (1 × 10^5^/well) which were pre-coated with a growth-factor-reduced Matrigel matrix (250 μL/well), followed by stimulation with Ang-1 (200 ng/mL) for 12 h. The tubes were then photographed and the total tube length in 5 fields was quantified using Image J software [37].

### 4.9. Western Blotting

HUVECs cultured in 6-well plates were allowed to grow to 90% confluence and exposed to MGO (100 μM) for 24 h, followed by stimulation with Ang-1 (200 ng/mL) for 15 min. In some experiments, the cells were pretreated with AG (100 μM) for 1 h before MGO treatment. The HUVECs were then lysed in RIPA buffer (Beyotime, Shanghai, China) supplemented with 1% (*v*/*v*) protease phosphatase inhibitor cocktail (Thermo Scientific). Equal samples were subjected to SDS-PAGE and transferred to polyvinylidene fluoride membranes (Bio-Rad Laboratories, Hercules, CA, USA). The membranes were blocked with rapid blocking buffer (EpiZyme, Shanghai, China) for 15 min at room temperature and incubated with primary antibodies against phosphorylated Tie-2 (1:1000), phosphorylated Akt (1:1000), total Tie-2 (1:1000) and total Akt overnight at 4 °C. The membranes were then incubated with species-specific horseradish peroxidase-conjugated goat IgG (Beyotime) for primary antibodies for 60 min at room temperature. After stripping, the same membranes were probed with anti-GAPDH (1:1000) or anti-β-actin (1:1000) antibodies. Chemiluminescence was used to visualize the protein bands, and the band density was measured with Image J software.

### 4.10. Immunoprecipitation

HUVECs were stimulated with MGO for 24 h and total protein samples were prepared. A classic immunoprecipitation kit was used to carry out protein immunoprecipitation [38]. Briefly, equal amounts of protein were incubated with protein A/G plus agarose beads and anti-Tie-2 antibody overnight at 4 °C with gentle rotation. According to the kit instructions, immune complexes were captured and solubilized by 50 μL SDS sample buffer. The immune complexes were then subjected to SDS-PAGE and immunoblotted with primary antibodies against AGEs (1:500) and Tie-2 (1:1000).

### 4.11. Ex Vivo Angiogenesis Assay

Mouse aortae isolated from ND-fed C57BL/6J mice were pretreated with AG (100 μM) or vehicle control for 1 h, followed by exposure to MGO (100 μM) for 24 h. The aortic rings were then embedded into collagen I gel and stimulated with Ang-1 (200 ng/mL) or vehicle control [38,39]. Vessel outgrowth at day 14 was determined with a Zeiss microscope and the area of sprouting vessels was quantified with Image J software.

### 4.12. Data Analysis

All of the data are presented as mean ± standard deviation (SD). An unpaired t-test and one-way ANOVA followed by post hoc comparison were carried out to analyze the results. All of the statistical analyses were performed using SPSS 19.0. A *p* value < 0.05 was considered significant.

## Figures and Tables

**Figure 1 ijms-23-07137-f001:**
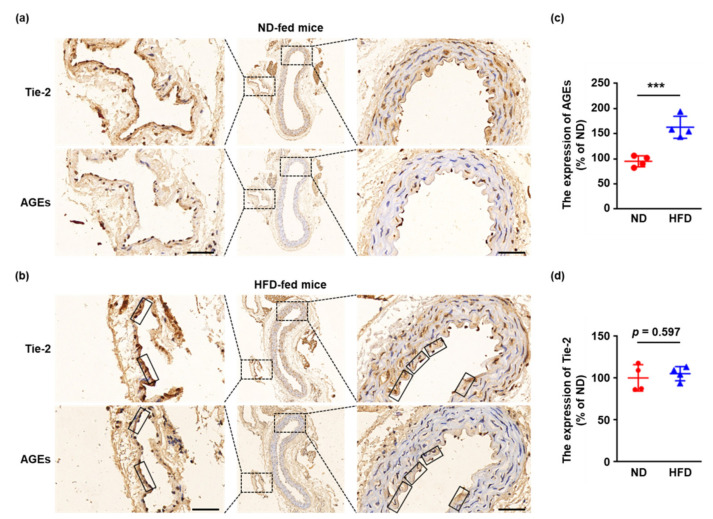
Tie-2 is glycated in HFD–fed mice. (**a**) The aortae derived from ND-fed C57BL/6J mice were isolated, fixed, embedded in paraffin, and sectioned. The Tie-2 and AGEs were then immunohistochemically stained in two serial aortic sections. Distance bars, 50 μm. Representative images are shown. *n* = 4. (**b**) The aortae derived from HFD-fed C57BL/6J mice were isolated, fixed, embedded in paraffin, and sectioned. The Tie-2 and AGEs were then immunohistochemically stained in two serial aortic sections. The solid black rectangles show the colocalization of Tie-2 and AGEs. Distance bars, 50 μm. Representative images are shown. *n* = 4. (**c**) Quantitative assessment of AGE expression was performed. (**d**) Quantitative assessment of Tie-2 expression was performed. *** *p* < 0.001. Error bar represents the standard deviation and *p* value was generated by *t* test.

**Figure 2 ijms-23-07137-f002:**
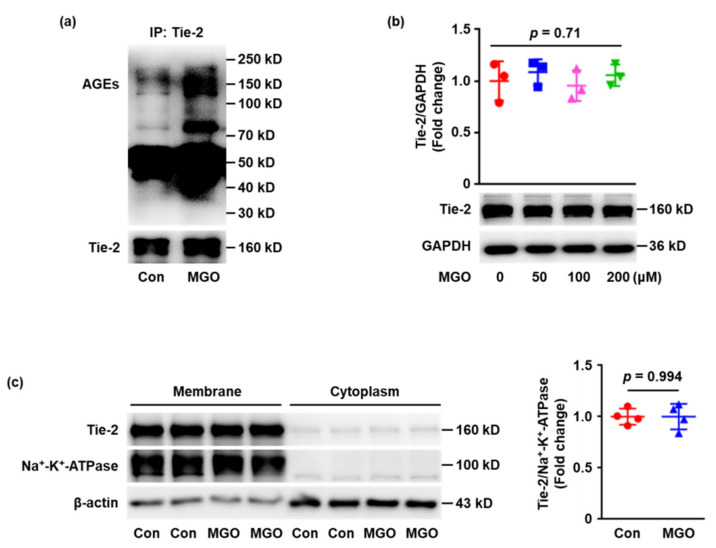
Tie-2 is glycated in MGO–treated endothelial cells. (**a**) HUVECs were exposed to MGO (100 μM) or vehicle control for 24 h. The cell lysates were immunoprecipitated with anti-Tie-2 antibody and the captured proteins were analyzed by Western blotting with anti-AGE and anti-Tie2 antibodies. Representative images of three independent experiments are shown. (**b**) HUVECs were stimulated with MGO (0, 50, 100 and 200 μM) for 24 h and total expression of Tie-2 was detected using Western blotting. Representative images of three independent experiments are shown and densitometric analysis of Tie-2 normalized to GAPDH was performed. (**c**) HUVECs were stimulated with MGO (100 μM) or vehicle for 24 h and the cell membrane proteins were separated. The Tie-2 expression at the cell membrane was determined with Western blotting and representative images of three independent experiments are shown. The densitometric analysis of Tie-2 normalized to Na^+^-K^+^-ATPase was carried out. Error bar represents the standard deviation and *p* value was generated by *t* test.

**Figure 3 ijms-23-07137-f003:**
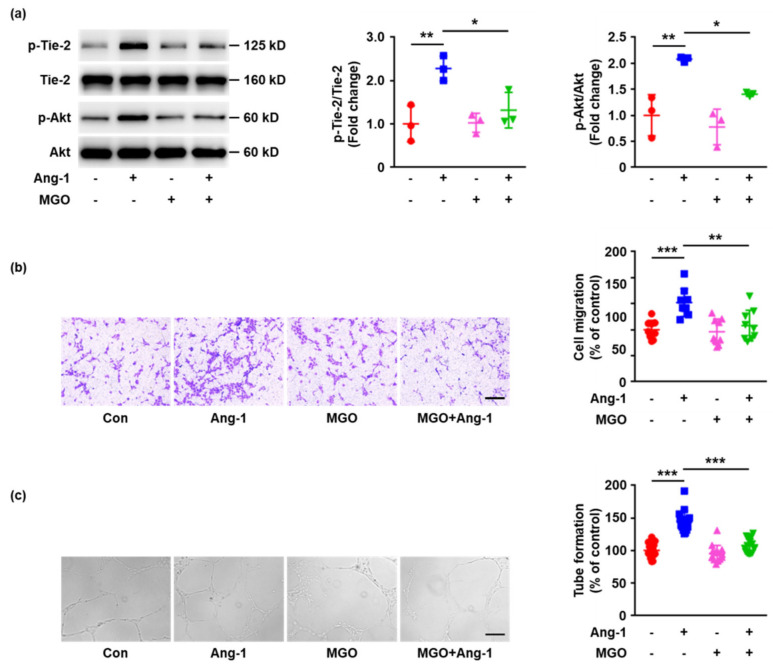
MGO inhibits Ang-1 signaling activation and Ang-1–induced angiogenesis. (**a**) HUVECs were exposed to MGO (100 μM) or vehicle control for 24 h, followed by stimulation with Ang-1 (200 ng/mL) for 15 min. Phosphorylation (p) of Tie-2 and Akt and total Tie-2 and Akt were detected by Western blotting. Representative images of three independent experiments are shown and densitometric analysis of phosphorylated Tie-2 and Akt normalized to total Tie-2 and Akt was performed. (**b**) HUVECs cultured in six-well plates were exposed to MGO (100 μM) or vehicle control for 24 h. The cells were then detached and added to the upper chambers containing porous filters, followed by stimulation with Ang-1 (200 ng/mL) or vehicle control. After 24 h, the cells were fixed, stained, and the cells that had migrated to the lower chambers were counted. Representative images of three independent experiments are shown and quantitative assessment was performed. (**c**) HUVECs cultured in six-well plates were exposed to MGO (100 μM) or vehicle control for 24 h. The cells were then detached and seeded onto Matrigel in the presence of Ang-1 (200 ng/mL) or vehicle control for 18 h. Representative images of three independent experiments are shown and quantitative assessment was performed. All data shown are mean ± SD and are expressed as % of control. Distance bars, 200 μm. * *p* < 0.05, ** *p* < 0.01, *** *p* < 0.001. Error bar represents the standard deviation and *p* value was generated by one-way ANOVA followed by post hoc comparison.

**Figure 4 ijms-23-07137-f004:**
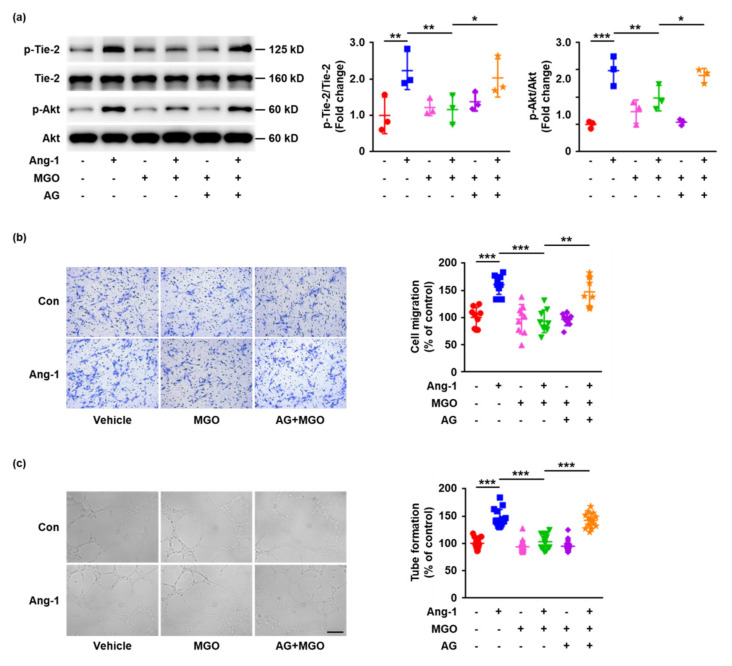
AG prevents MGO–induced Ang-1/Tie-2 signaling dysfunction. (**a**) HUVECs were pre-treated with AG (100 μM) or vehicle control for 1 h and then exposed to MGO (100 μM) for 24 h, followed by stimulation with Ang-1 (200 ng/mL) or vehicle control for 15 min. Phosphorylation (p) of Tie-2 and Akt and total Tie-2 and Akt were detected by Western blotting. Representative images of three independent experiments are shown and densitometric analysis of phosphorylated Tie-2 and Akt normalized to total Tie-2 and Akt was performed. (**b**) HUVECs cultured in six-well plates were pre-treated with AG (100 μM) or vehicle control for 1 h, followed by exposure to MGO (100 μM) for 24 h. The cells were then detached and added to the upper chambers containing porous filters, followed by stimulation with Ang-1 (200 ng/mL) or vehicle control. After 24 h, the cells were fixed, stained, and the cells that had migrated to the lower chambers were counted. Representative images of three independent experiments are shown and quantitative assessment was performed. Distance bars, 200 μm. (**c**) HUVECs cultured in six-well plates were pre-treated with AG (100 μM) or vehicle control for 1 h, followed by exposure to MGO (100 μM) for 24 h. The cells were then detached and seeded onto Matrigel in the presence of Ang-1 (200 ng/mL) or vehicle control for 18 h. Representative images of three independent experiments are shown and the quantitative assessment was performed. Distance bars, 200 μm. * *p* < 0.05, ** *p* < 0.01, *** *p* < 0.001. Error bar represents the standard deviation and *p* value was generated by one-way ANOVA followed by post hoc comparison.

**Figure 5 ijms-23-07137-f005:**
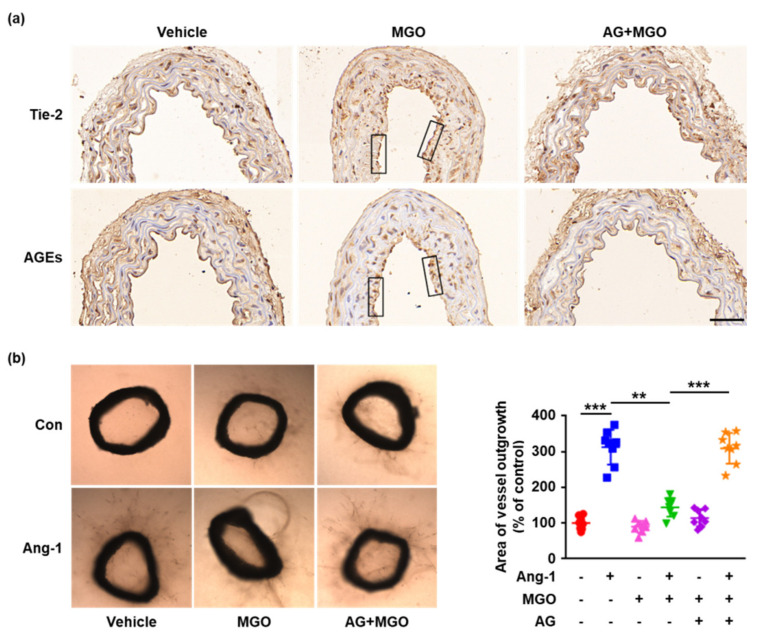
Glycation of Tie-2 inhibits Ang-1–induced vessel outgrowth ex vivo. The aortae derived from ND-fed C57BL/6J mice were isolated, treated with AG (100 μM) or vehicle control for 1 h, followed by exposure to MGO (100 μM) for 24 h. (**a**) The aortae were then fixed, embedded in paraffin, and sectioned. The Tie-2 and AGEs were immunohistochemically stained in two serial aortic sections. The solid black rectangles show the colocalization of Tie-2 and AGEs. Distance bars, 50 μm. Representative images are shown. *n* ≥ 6 per group. (**b**) The aortic rings were embedded into collagen I gel and stimulated with Ang-1 (200 ng/mL) or vehicle control for 14 d. Representative images are shown. *n* = 8 per group. Quantitative analysis of microvessel sprouts from the aortic rings was carried out. ** *p* < 0.01, *** *p* < 0.001. Error bar represents the standard deviation and *p* value was generated by one-way ANOVA followed by post hoc comparison.

**Figure 6 ijms-23-07137-f006:**
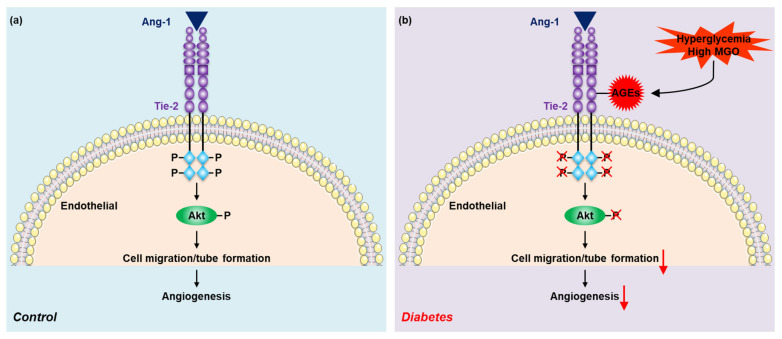
The molecular mechanism underlying Tie-2 glycation–regulated Ang-1 signaling and Ang-1–induced angiogenesis. (**a**) Under the physiology condition, the binding of Ang-1 to Tie-2 induces Tie-2 and Akt activation, causing endothelial cell migration and tube formation, consequently leading to angiogenesis. (**b**) However, hyperglycemia or high MGO in diabetes mellitus induces AGEs–Tie-2 adduct formation (Tie-2 glycation), which inhibits Ang-1 signaling activation and Ang-1-induced angiogenesis.

## Data Availability

The data presented in this study are available on request from the corresponding author.

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
