# Peer review of "Glycation of Tie-2 Inhibits Angiopoietin-1 Signaling Activation and Angiopoietin-1-Induced Angiogenesis"

_ijms, 2022, doi:10.3390/ijms23137137_

Round 1

Reviewer 1 Report

Zhou and colleagues demonstrate in vivo, in a mouse model treated with a high-fat diet, HFD, that Tie2 receptor expression colocalizes with that of AGEs, and in MGO-treated aorta rings, Ang1-mediated sprouting is markedly reduced. In vitro, on HUVECs, treatment with MGO promotes glycation of Tie2, its inactivation, and consequently, inhibits Ang1-mediated proangiogenic functions. The glycation inhibitor AG reverses both on aorta rings and in vitro, on HUVEC, the activity of MGO on Ang1-mediated Tie2 signaling.

The data are very interesting for potential therapeutic implications in the clinical setting in diabetes, and the manuscript is well written and articulate.

Comments:

In vitro and ex vivo, the authors use 100 µM MGO. In the discussion, the authors stated that this metabolite is increased in the serum of diabetic patients. Is this concentration, 100 µM, possible? That is, is it likely that in the circulation or perivascular tissues of patients with diabetes, an MGO concentration of 100 µM is achieved?

How does AG work? It is important that the authors describe the mechanism of action of AG and justify the exposure of cells and aorta rings to AG 1 hour before treatment with MGO.

Because the authors are interested in diabetes, it is important for the authors to demonstrate that HFD treatment promotes increased blood glucose in mice.

Please, report the origin of the type I collagen used.

In the results, the authors state that the mice were treated 16 weeks on the high-fat diet, while in Materials and Methods they report 20 weeks; please correct where necessary.

Please, report what is the vehicle/control for MGO in materials and methods

In materials and methods, correct “digested cells” with “detached cells”

In all materials and methods, in vitro, serum conditions are missing and should be reported.

Finally, it is necessary for the authors to explain why they do not show the response of aortas isolated from HFD-treated mice to Ang1, but use only aortas isolated from ND-treated mice for ex vivo experiments.

Reviewer 2 Report

Zhou et al present a paper describing the role of glycation of Tie-2 in the inhibition angp-1 signaling and explain its mechanism. 

I found the paper solid, with good merit and a medium-to-high level of novelty. The used language is fine, albeit, please avoid mixing American and British English. 

The research is fairly designed, however, is based solely on histological and western blotting data, which makes it a little bit less convincing. The obtained results are in line with general trends in the area and at the same point fill some gaps in the knowledge. 

Indeed, I do not have many majors, just one and some minors: 

The major - why do not try to perform some RT-PCR experiments to make the conclusions a bit more stronger? Observing phenomena during two parallell experiments - westerns and pcrs will bring more novelty and reliability to the paper. 

Minor:

Please provide conditions at which animals were kept as well as the number of Animal Committee approval

Please provide catalog numbers of used antibodies

Did the Authors start to feed animals with an HFD diet 8 weeks after their birth? Why not earlier? Is any reference to support this workflow? 

Which antibiotic has been used for the cell culture experiment? 

Figure 3B and C as well 4B and C - please provide the Readers with the arrows pointing out the point of the interest

Please reconsider expanding the keyword section

What statistical software has been used? 

Round 2

Reviewer 2 Report

The Authors correctly addressed my concerns as well as explained all the doubts.

Right now, I would like to see only two minors introduced, after that, the paper will fit the journal scope well:

1) Please expand the limitation (line 298) section by adding the information that the additional approach to confirm the finding should be performed.

2) Since the Authors emphasize the importance of  AGE-related glycation it would be nice to highlight within 2-3 sentences recent advances in the pharmacological approach to prevent/treat it. These recently published papers would be handy: 

https://doi.org/10.3390/medicina58040472

DOI: 10.2174/1381612822666161006143032

Once again, I am happy with the revision process. 

  •  
